# Fine-scale family structure shapes influenza transmission risk in households: Insights from primary schools in Matsumoto city, 2014/15

**Akira Endo** [1]*, **Mitsuo Uchida**[2], **Adam J. Kucharski**[1,3], **Sebastian Funk**[1,3]

**1** Department of Infectious Disease Epidemiology, London School of Hygiene & Tropical Medicine, London, United Kingdom, **2** Department of Public Health, Graduate School of Medicine, Gunma University, Gunma, Japan, **3** Centre for the Mathematical Modelling of Infectious Diseases, London School of Hygiene and Tropical Medicine, London, United Kingdom

\* akira.endo@lshtm.ac.uk

**Data Availability Statement:** All data are available from the Github repository (https://github.com/akira-endo/HHstudy_FluMatsumoto2014-15).

## Abstract

Households are important settings for the transmission of seasonal influenza. Previous studies found that the per-person risk of within-household transmission decreases with household size. However, more detailed heterogeneities driven by household composition and contact patterns have not been studied. We employed a mathematical model that accounts for infections both from outside and within the household. The model was applied to citywide primary school seasonal influenza surveillance and household surveys from 10,486 students during the 2014/15 season in Matsumoto city, Japan. We compared a range of models to estimate the structure of household transmission and found that familial relationship and household composition strongly influenced the transmission patterns of seasonal influenza in households. Children had a substantially high risk of infection from outside the household (up to 20%) compared with adults (1–3%). Intense transmission was observed within-generation (between children/parents/grandparents) and also between mother and child, with transmission risks typically ranging from 5–20% depending on the transmission route and household composition. Children were identified as the largest source of secondary transmission, with family structure influencing infection risk.

## Author summary

We characterised detailed heterogeneity in household transmission patterns of influenza by applying a mathematical model to citywide primary school influenza survey data from 10,486 students in Matsumoto city, Japan, one of the largest-scale household surveys on seasonal influenza. Children were identified as the largest source of secondary transmission, with family structure influencing infection risk. This suggests that vaccinating children would have stronger secondary effects on transmission than would be assumed without taking into account transmission patterns within the household.

**Funding:** AE receives financial support from The Nakajima Foundation (http://www.nakajimafound.or.jp/). AJK [206250/Z/17/Z] and SF [210758/Z/18/Z] are sponsored by the Wellcome Trust (https://wellcome.ac.uk/). The funders had no role in study design, data collection and analysis, decision to publish, or preparation of the manuscript.

## Introduction

Respiratory infectious diseases transmitted by droplets, exemplified by influenza, are known to spread through social contact networks [1,2]. Social settings that involve frequent contacts play important roles in transmission dynamics [3,4]. Households are considered one of the main layers of transmission because individuals come in close contact with each other both conversationally and physically on a daily basis [5–8]. Several epidemiological studies have used household data to investigate the transmission dynamics of influenza within households [9,10], particularly in terms of the secondary attack rate (the number of secondary household cases divided by the number of household members at risk). However, this assumes that an index case (the first case in a household, who is considered to be infected outside the household) is responsible for all subsequent household cases and that all the other household members are equally at the risk of secondary infection.

The possibility of co-primary infections and tertiary transmission is neglected under such assumptions [9], with potentially heterogeneous transmission patterns between household members also radically simplified. The former limitation can be addressed by mathematical models which separately estimate the risk of infection from outside the household and the within-household transmission risk [11]. Many household studies have employed the Longini-Koopman model and other related models to study the within-household transmission dynamics of influenza [12–18].

On the other hand, potentially-heterogeneous transmission patterns have not been fully studied with empirical data. Multiple household modelling studies have incorporated factors including age, vaccination status and antibody titres to account for heterogeneity, but these are usually used to identify individual risk factors that determine relative susceptibility of individuals [15,17,19–21]. Given typical behaviours within the family, it is natural to expect substantial heterogeneity in household contact patterns related to familial relationships and household compositions, on top of those individual factors [7,8]. Addy et al. [22] estimated a within-household transmission matrix consisting of two classes (children and adults), but more classes might be needed to better account for the heterogeneity of household contact patterns. In actual implementation, even such two-class analysis is very rare; in most cases, household size is the only family-related covariate for modelling of within-household transmissions in outbreak data [14,15,18,19,23]. Further, due to the limited sample size of households in these studies, a rationale on the quantitative effect of household size in transmission has not been established. Familial roles/relationships (e.g., father, mother, grandparent, etc.) have been paid far less attention to in household outbreak studies; we found only one field study on influenza that included familial roles as a covariate, a descriptive study that did not quantify the risk by familial roles [24].

Households serve as important units in intervention policies [25,26]. Tailored quantification of the transmission risks from outside and inside the household could help prioritise and promote household-level prevention strategies including vaccination. If specific compositions of households have a higher risk of outbreaks than others, intervention policies may be optimised by targeting such households. Moreover, as vaccine uptake is shown to be influenced by the perceived risk of infection and vaccine effectiveness [27,28], identifying the household-specific risk of infection and the possible reduction by vaccines may support highlight the individual benefit of vaccination.

To investigate the within-household transmission dynamics of seasonal influenza, we applied a highly flexible household transmission model that accounts for heterogeneity to a large influenza dataset. The dataset included more than 10,000 primary school students with the infection status not only of students but also of their household members, which enabled a detailed investigation of within-household transmission dynamics. Focusing on the effect of

familial roles and household compositions, we compared multiple models with different levels of complexity to find the best model to describe the transmission patterns.

## Methods

### Ethics statement

The data analysis in the present study was secondary and was approved by the ethics committee at the London School of Hygiene & Tropical Medicine (reference number: 14599). Consent was not obtained because the data were anonymous upon collection. The original study was approved by the Committee for Medical Ethics of Shinshu University (reference number: 2715).

### Data source

We used data from a citywide primary school influenza survey. At the end of the 2014/15 season (early March), parents of students at all 29 public primary schools in Matsumoto city, Nagano prefecture, Japan, were asked to respond to a questionnaire consisting of a variety of questions including whether the students had influenza during the season, onset date and observed symptoms, vaccination history, family composition and who in the same household had influenza episodes during the season. The data was originally collected for an observational study on the effect of prevention measures against seasonal influenza (Uchida et al., 2017) [29]. In the present study, we only considered data on influenza episodes in students, their household composition and influenza episodes in the household members. Participants reported the number of siblings in the household, and also ticked the type of family members (such as "father", "younger sister" or "uncle") with whom they live, as well as whether they acquired influenza in the 2014/15 season. Among 13,217 students eligible, 11,390 (86%) responded to the survey. After removing those with missing values, 10,486 surveys were used in the present study. Characteristics of the population and frequent household compositions are shown in Tables 1 and 2. The influenza types reported for the student cases during the season were mostly A (95% of those tested positive) [30]. The national-level surveillance data suggested that AH3 strain was predominant, accounting for 99% of the type A isolates [31]. The

**Table 1. The number of individuals and influenza cases in each type.**

| Individual type | | | Counts* | Cases* | Attack ratio (%) |
|---|---|---|---|---|---|
| Student | Overall | | 10,410 | 2,137 | 20.5 |
| | Male | | 5,311 | 1,132 | 21.3 |
| | Female | | 5,099 | 1,005 | 19.7 |
| | | Grade 1 | 1,831 | 406 | 22.2 |
| | | 2 | 1,773 | 363 | 20.5 |
| | | 3 | 1,731 | 342 | 19.8 |
| | | 4 | 1,717 | 375 | 21.8 |
| | | 5 | 1,674 | 322 | 19.2 |
| | | 6 | 1,684 | 329 | 19.5 |
| Father | | | 9,201 | 629 | 6.8 |
| Mother | | | 10,260 | 866 | 8.4 |
| Sibling | | | 12,632 | 2,320 | 18.4 |
| Other | | | 4,356 | 191 | 4.4 |

* The number of respondents and cases for "Father", "Mother", "Sibling" and "Other" is obtained from the response to the questionnaire and may be redundant due to the inclusion of multiple students from the same household.

**Table 2. Frequency distribution table for compositions of households included in the retrospective data.**

| Order | Composition | # of households | Order | Composition | # of households |
|---|---|---|---|---|---|
| 1 | FM-2 | 3,915 | 11 | M-3 | 160 |
| 2 | FM-3 | 1,971 | 12 | FM-1-2 | 134 |
| 3 | FM-1 | 899 | 13 | FM-1-1 | 97 |
| 4 | FM-2-2 | 606 | 14 | M-1-2 | 86 |
| 5 | M-2 | 429 | 15 | M-2-2 | 80 |
| 6 | FM-2-1 | 415 | 16 | FM-2-3 | 70 |
| 7 | FM-3-2 | 297 | 17 | FM-3-3 | 57 |
| 8 | FM-4 | 250 | 18 | FM-4-2 | 55 |
| 9 | FM-3-1 | 232 | 19 | M-1-1 | 43 |
| 10 | M-1 | 205 | 20 | M-2-1 | 39 |
| | | | | Subtotal | 10,040 (95.7%) |

Only 20 most frequent compositions are shown, accounting for 95.7% of the total 10,486 responses. Household compositions are denoted in the following manner. FM: households with both father and mother; M: households with a single mother; The first number: the total number of siblings in the household; The second number (where applicable): the number of other members (mostly grandparents) in the household.

vaccination coverage of the students in the dataset was 48%; however, we did not consider vaccination in the present analysis. Further details of the data collection and descriptive epidemiology can be found in the original studies [29,30].

In the present study, we classified each individual in households as one the following type: "father", "mother", "student", "sibling", or "other". "Students" are participants of the survey (i.e., students of primary schools in Matsumoto city), and "siblings" are their elder/younger siblings, who may have also been recruited in the survey if they are primary school students (however, they are not linked in the data and thus unidentifiable as participants). The parameters for "students" and "siblings" were differentiated because "siblings" are not necessarily primary school students, therefore their characteristics may be different from "student". "Father" and "mother" were labelled as "single-parent" if they are only one parent in the family; models were considered in the model selection where their parameter values were differentiated from cohabiting parents (details described in "model selection"). Most individuals classified as "other" were grandparents (90.1%). Uncles/aunts accounted for 6.7%, and the remaining 3.2% was "none of the above categories".

In the survey, all students who were reported to have acquired influenza were also reported to have been diagnosed at a medical institution. For other household members, clinical diagnosis was not clearly required on the question sheet. In Japan, rapid diagnostic tests are usually used for suspected patients. International systematic reviews estimated that the sensitivity and specificity of rapid diagnostic tests are 50–70% and 98–99%, respectively [32,33]. However, the sensitivity for studies conducted in Japan included in these reviews was relatively high (range: 72.9–96.4%), consistent with other earlier studies conducted in Japan [34–36]. Considering that many Japanese primary schools encourage students presenting influenza-like symptoms to consult medical institutions so that they are granted absence, we believe that the reported influenza episodes in the dataset were sufficiently inclusive for our analysis. We also performed a sensitivity analysis to address possible underreporting in the survey (described later).

### Heterogeneous chain binomial model

We employed the chain-binomial model presented in [37] which allows for heterogeneous transmission (see Fig 1 for schematic illustration). Let *N* be a vector representing the number

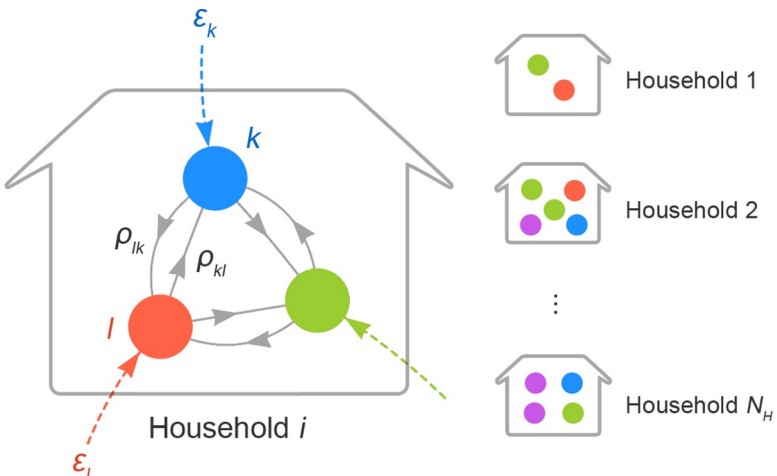

**Fig 1. A schematic illustration of household chain-binomial model.** Nodes in different colours correspond to different types of individuals (e.g., father, sibling, etc.). Transmission patterns are illustrated taking household $i$ as an example. Coloured dotted edges represent the risk of external infection $\boldsymbol{\varepsilon}$ to each individual. Solid grey edges denote person-to-person transmission risk (PTR) from one type of person to another. PTR from type $l$ to $k$ is given as $\rho_{kl}$, which refers to the risk of transmission given that the individual of type $l$ is infectious. Households have different compositions and $\rho_{kl}$ may also vary according to the composition. On the other hand, $\boldsymbol{\varepsilon}$ is the risk from outside the household and thus assumed to be identical across households.

of family members stratified by individual type (e.g., father, mother, child, etc.) in a household. The probability that a certain combination of individuals (represented by a vector $\boldsymbol{n}$) in the household are infected by the end of the season is given by the following recursive equations.

$$\pi(\boldsymbol{n}; \boldsymbol{N}, \boldsymbol{\varepsilon}, H) = \pi(\boldsymbol{n}; \boldsymbol{n}, \boldsymbol{\varepsilon}, H) \prod_k \binom{N_k}{n_k} S_k(\boldsymbol{n}, \boldsymbol{\varepsilon}, H)^{N_k - n_k}, \tag{1}$$

$$\pi(\boldsymbol{n}; \boldsymbol{n}, \boldsymbol{\varepsilon}, H) = 1 - \sum_{\boldsymbol{v} < \boldsymbol{n}} \pi(\boldsymbol{v}; \boldsymbol{n}, \boldsymbol{\varepsilon}, H).$$

where $N_k$ and $n_k$ are the $k$-th component of $\boldsymbol{N}$ and $\boldsymbol{n}$, respectively ($1 \leq k \leq K$). The sum $\sum_{\boldsymbol{v} < \boldsymbol{n}}$ is taken for all vector $\boldsymbol{v}$ satisfying $0 \leq v_k \leq n_k$ ($\forall k$) and $\boldsymbol{v} \neq \boldsymbol{n}$. We denoted by $\boldsymbol{\varepsilon}$ the external risk of infection over the season for each type of individual. The susceptible-infectious transmission probability (SITP) $\rho_{kl}$ is the probability of within-household transmission for a specific infectious-susceptible pair [18] and has been used to quantify within-household transmission. However, it is more convenient to use the effective household contact matrix $H = (\eta_{kl})$ in the model; $\eta_{kl}$ is defined to satisfy $\rho_{kl} = 1 - \exp(-\eta_{kl})$, and is interpreted as the amount of contact that leads to within-household transmission (effective contact) from type $l$ to $k$. That is, $\eta_{kl}$ denotes the amount of exposure that an individual $k$ experiences when another individual of type $l$ in the same household is infectious. $S_k(\boldsymbol{n}, \boldsymbol{\varepsilon})$, the probability that a type $k$ individual escapes infection from both outside and inside the household throughout the season, is given as

$$S_k(\boldsymbol{n}, \boldsymbol{\varepsilon}, H) = (1 - \varepsilon_k) \exp(-\sum_l \eta_{kl} n_l). \tag{2}$$

$(1 - \varepsilon_k)$ is the probability that the individual is not infected outside the household, and $\exp(-\sum_l \eta_{kl} n_l)$ is the probability that the individual is not infected from any of the household infectives. When a dataset $\{\boldsymbol{N_i}, \boldsymbol{n_i}\}$ contains the family composition and infection status in each household $i$, the pseudo-likelihood function (where interaction between households is

neglected) is given as

$$L(\boldsymbol{\varepsilon}, H; \{\boldsymbol{N_i}, \boldsymbol{n_i}\}) = \prod_i \pi(\boldsymbol{n_i}; \boldsymbol{N_i}, \boldsymbol{\varepsilon}, H). \tag{3}$$

The household-wise likelihood $\pi(\boldsymbol{n_i}; \boldsymbol{N_i}, \boldsymbol{\varepsilon}, H)$ is computed by recursively applying Eq (1) starting with $\pi(\boldsymbol{0}; \boldsymbol{0}, \boldsymbol{\varepsilon}, H) = 1$.

## Transmission risk in households

We modelled the possible heterogeneity in household transmission by parameterising the effective household contact matrix $H = (\eta_{kl})$. Our basic assumptions are: (i) each pair of individuals have a specific "intensity of contact"; (ii) the relative importance of each household contact may be reduced if an individual experiences a large amount of household contacts in total; (iii) the contact intensity adjusted by the total amount of contact is proportional to the force of infection. That is, we modelled $\eta_{kl}$ as

$$\eta_{kl} = \beta \frac{c_{kl}}{C_k^{\gamma}}. \tag{4}$$

The contact intensity $c_{kl}$ represents the (hypothetical) number of household contacts between type $k$ and $l$, and $\beta$ is the transmissibility coefficient. $C_k$ represents the total number of household contacts experienced by an individual of type $k$, which we introduced to investigate how $\eta_{kl}$ differs in households of different sizes and compositions. Noting that the number of individuals in contact is $N_k - 1$ if $k = l$, we get

$$C_k = \sum_l c_{kl}(N_l - \delta_{kl}), \tag{5}$$

where $\delta_{kl}$ is the Kronecker delta. The value of the exponent parameter $\gamma$ determines how strongly $\eta_{kl}$ is scaled by $C_k$, which associates our model with density-dependent vs. frequency-dependent mixing assumptions [38]. The value $\gamma = 0$ corresponds to the density-dependent mixing assumption, where the force of infection is proportional to the total number of contacts (weighted by intensity) with infectives, whereas $\gamma = 1$ corresponds to the frequency-dependent mixing assumption, where it is the proportion of infectious contacts among total contacts that matters. In addition to $\gamma = 0$ and $\gamma = 1$, $\gamma$ was also allowed to be estimated as a free parameter in the model selection, representing a mixture of density-dependent and frequency-dependent mixing.

The contact intensity matrix $(c_{kl})$ is interpreted as the per-individual version of the contact matrix ($c_{kl} = b_{kl}/N_l$ where $b_{kl}$ is the contact matrix). The parameter $c_{kl}$ generally constitutes a $K \times K$ matrix and contains too many parameters to estimate. We, therefore, reduced the number of parameters by categorising contacts into the following 5 pairs first:

$$c_{kl} = \begin{cases} c_{CC} \ (\text{Child} - \text{Child}) \\ c_{FC} \ (\text{Father} - \text{Child}) \\ c_{MC} \ (\text{Mother} - \text{Child}) \\ c_{OC} \ (\text{Other} - \text{Child}) \\ c_{AA} (\text{Adult} - \text{Adult}) \end{cases} \tag{6}$$

Child included both "student" and "sibling", and adult included "father", "mother" and "other". (In models where "single-parent" is a separate type, another parameter $c_{SC}$ (Single parent–Child) was added.) The matrix was assumed to be symmetric, i.e, $c_{kl} = c_{lk}$. We did not vary $\beta$ in our baseline analysis such that transmission is also symmetric ($\eta_{kl} = \eta_{lk}$), but the possibility of type-specific susceptibility was addressed in our sensitivity analysis. Since we did not have a measurement for the intensity of household contacts in our dataset, we used relative values of

$c_{kl}$ in our analysis where $c_{AA}$ was assumed to be 1. The parameter $\beta$ is approximately equal to the probability of transmission in a (hypothetical) household composed of only father and mother (since $\frac{c_{kl}}{C_k^i} = 1$ regardless of $\gamma$).

## Statistical analysis and model selection

We sampled parameter values from a posterior distribution yielded from the pseudo-likelihood function (3) and priors in Table 3 using the Markov-chain Monte Carlo (MCMC) method. An optimal variance-covariance matrix for proposal was explored by the adaptive Metropolis algorithm, and then the random-walk Metropolis algorithm was used to obtain final samples. All MCMC sampling was performed using the R package {LaplacesDemon}. The scripts and dataset to produce MCMC samples for the main results are reposited on GitHub (https://github.com/akira-endo/HHstudy_FluMatsumoto2014-15).

First, we tested various possible combinations of assumptions on the effective contact matrix and the risk of external infection (shown in Table 3) and compared their goodness of fit by Widely-applicable Bayesian Information Criterion (WBIC) [39]. Model variants included (i) homogeneous or heterogeneous mixing in households ($c_{kl}$), (ii) uniform or heterogeneous risk of external infection ($\varepsilon_k$), (iii) the value of the exponent parameter ($\gamma$), and (iv) whether the parameter values for a single parent is differentiated from those of cohabiting parents. Characteristics of compared models are documented in S1 File, Section 1. WBIC for each model was computed from 80,000 MCMC samples which were thinned from 125,000 samples × 8 chains so that the chains had the effective sample size (ESS) ~40,000.

We then used the models selected by WBIC to estimate the parameters. As final samples, 10,000 thinned samples were recorded from 40,000 pre-thinned MCMC samples. It was ensured that the ESS was at least 500 for each parameter.

Using the estimated parameters, we computed the source-stratified risk of infection and the risk attributable to the introduction into the household (see S1 File, Section 2 for further details).

## Further model development

When the parameters were estimated with the best model selected, we found that the estimates for $c_{FC}$ and $c_{OC}$ were very similar, which suggested that we might be able to equate these two

**Table 3. Parameter estimates by the best model.**

| Parameter | | Prior | Estimate (95% CrI) |
|---|---|---|---|
| External risk ($\varepsilon_k$) | Student | 1-LogUnif(0,1)* | 0.197 (0.188–0.207) |
| | Sibling | | 0.161 (0.153–0.169) |
| | Mother | | 0.035 (0.030–0.040) |
| | Father | | 0.038 (0.033–0.043) |
| | Other | | 0.013 (0.009–0.017) |
| Contact intensity ($c_{kl}$) | Child-Child | Unif(0,∞) | 1.04 (0.88–1.23) |
| | Mother-Child | | 1.16 (1.00–1.32) |
| | Father-Mother | | 1 (0.748–1.282) |
| | Other-Other | | 1.97 (1.10–3.24) |
| | Cross generational | | 0.43 (0.35–0.52) |
| Transmissibility ($\beta$) | | (derived quantity; not sampled by MCMC) | 0.20 (0.16–0.24) |
| Exponent parameter ($\gamma$) | | Unif(−∞,∞) | 0.51 (0.33–0.69) |

* Cumulative force of infection is uniformly distributed.

parameters and further stratify the contacts between adults ($c_{AA}$) with the degree of freedom earned. We tested some other contact intensity matrices, including

$$
c_{kl} = \begin{cases}
c_{CC} \ (\text{Child} - \text{Child}) \\
c_{MC} \ (\text{Mother} - \text{Child}) \\
c_{FM} \ (\text{Father} - \text{Mother}) \\
c_{OO} \ (\text{Other} - \text{Other}) \\
c_{X} \ (\text{Cross generational})
\end{cases}
\tag{7}
$$

which gave the best performance in the end. Explored candidate models and selection results are detailed in S1 File, Section 2.

## Sensitivity analysis

We performed a sensitivity analysis to address potential biases in our dataset. We considered in our sensitivity analysis (i) ascertainment bias, (ii) different susceptibility in children, (iii) multiple counting of households and (iv) censoring of sibling cases.

The first two points are related to the assumptions in our models. Influenza can have a low reporting rate due to mild clinical presentation (including asymptomatic infections), and therefore some infectious individuals may not have been included in our dataset. The reporting rate of influenza is considered to be very high in primary school students in Japan, who are often required to report influenza to their schools. On the other hand, the reporting rate of adults can be lower, as they may be less likely to seek medical treatment than children. A sero-survey conducted in Japan after the 2009/10 H1N1 influenza pandemic suggested that while influenza in children was almost fully reported, the reporting rate of adults were relatively low (30–50%) [40].

Another possible difference between adults and children is susceptibility: adults may be less likely to be infected by the same amount of exposure due to the previous history of infections or stronger immune systems than children. Conversely, children may exhibit lower susceptibility if the vaccine uptake for them is higher than adults. The majority of household transmission studies from a systematic review [9] reported a significant association between susceptibility and age (although this becomes the minority when limited to the studies with PCR-confirmed cases). Our baseline model assumes that transmissibility $\beta$ is identical between individuals, but in reality, transmissibility might depend on the age of the susceptibles.

The remaining points explored in sensitivity analysis are inherent limitations in our dataset. One of the limitations is that, because students in the same household responded to the questionnaire separately, households with multiple siblings may have been counted more than once. As this was an anonymous questionnaire, data obtained from different students were not linked with each other even if they were from the same household. If there was more than one child in a household who was eligible for the study, the same household transmissions can appear multiple times in the dataset, which could modify the results. Lastly, because of the design of the questionnaire, the number of influenza cases in siblings may have been underreported. The questionnaire asked whether each type of individual in the same household had influenza during the season, and the respondents ticked if at least one individual of that type was infected since it was a yes-no question. Therefore, even if there was more than one case in the same type of individuals, the number was not reported and treated as a single case; that is, if a respondent has two older brothers, he/she only reports that "older brother had influenza", and there was no distinction on the dataset whether it was only one or both of them. This issue was addressed by modifying the likelihood function.

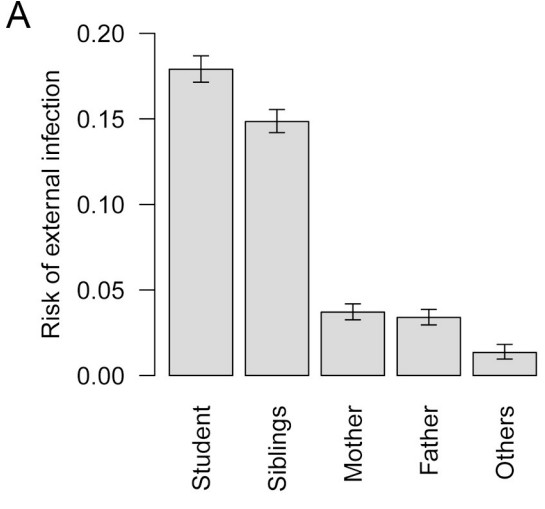
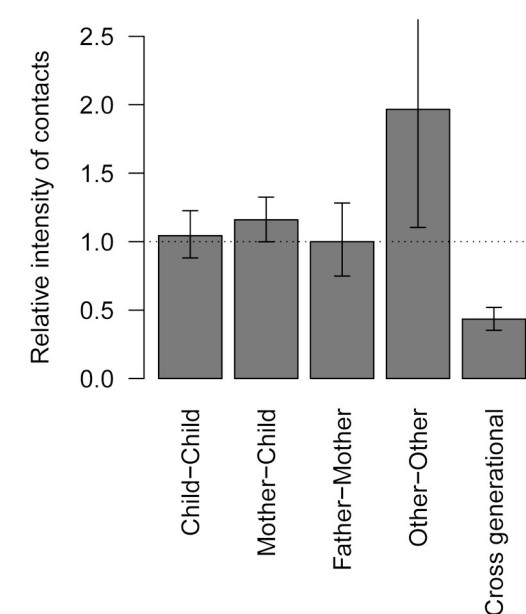

**Fig 2. Estimated risk of external infection and relative intensity of within-household contact.** (A) Estimated risk of external infection for each type of individual. (B) The estimated relative intensity of within-household contact. Values are scaled so that the median of contact intensity between adults is 1 (horizontal dotted line). Whiskers indicate 95% credible intervals (CrI).

Each potential source of bias was addressed by incorporating the data-generating process causing the bias into the model. Technical details of the sensitivity analysis can be found in S1 File, Section 3.

## Results

We found considerable heterogeneity in both the risk of external infection and the risk of within-household transmission (Table 3 and Fig 2). The best performing mathematical model suggested that children had a comparatively high risk of infection outside the household: 20% in the primary school students and 16% in their siblings, compared to only 1–3% in adults. Within-household contact patterns showed strong generational clustering. High contact intensities were observed within the same generation (between siblings, parents and grandparents), and the intensity of cross-generational contacts was less than half the intensity within the same generation. Contact between mothers and children was an exception to this, showing a higher intensity than between parents. The estimated contact intensity relative to that between parents (father-mother) was highest between other-other (1.97; CrI: 1.10–3.24), most of whom were grandparents in our data, followed by mother-child (1.16; CrI: 1.00–1.32) and child-child (1.04; 0.88–1.23), both of which are insignificantly different from father-mother (1; 0.75–1.28). The model did not support a significant difference between parameter estimates for single and cohabiting parents.

The inferred networks of household transmission suggest that various contact patterns between household members exist in different household compositions. The contact intensity between individuals are shown in network graphs (Fig 3A–3C) for three selected characteristic household composition models, "nuclear family": FM-2 (see Table 2 for the notation), (b) "many-siblings family": FM-4, and (c) "three-generation family": FM-2-2. Mothers served to bridge between the generations of children and parents; clusters of grandparents were relatively independent of other household members.

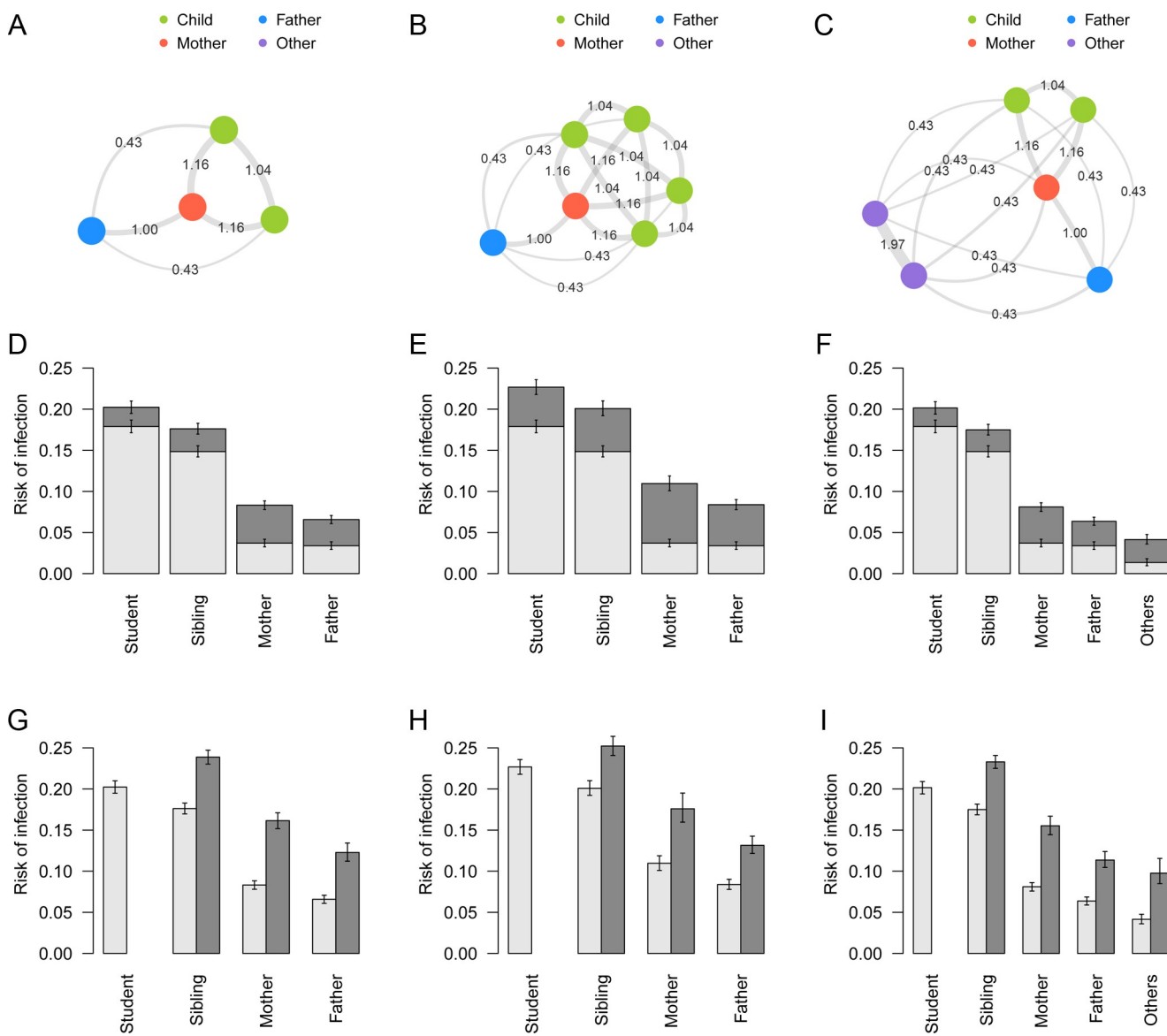

**Fig 3. Contact patterns and risk of infection in specific household compositions.** (A)-(C) Network graphs showing contact intensity between individuals for different household compositions: (A) "nuclear family", (B) "many-siblings family", (C) "three-generation family". Node colours represent the type of individuals. Edges denote the relative intensity of contact ($c_{kl}$) between individuals. (D)-(F) Risk of infection in households of different compositions stratified by source. Light grey: risk of infection from outside the household; dark grey: risk of infection from within the household. Whiskers indicate the 95% CrI. (G)-(I) Unconditional risk of infection and conditional risk given an introduction of infection into a household. Light grey: overall risk of infection for each individual in the household; dark grey: risk of overall infection conditional that a student is infected outside and introduces infection into the household. Infection of the student is considered given, and thus the conditional risk for the student is not shown. Whiskers indicate the 95% CrI.

The overall risk of infection and the breakdown of infection source presented in Fig 3D–3F suggests that risk of infection in children was mostly from outside the household, whereas a larger proportion of risk in adults was attributed to within-household transmission. Risk of within-household infection increased when more children were in the household (Fig 3E); however, the influence of additional members categorised as "others" (grandparents in most cases) was minimal, probably due to their low risk of external infection and contact intensity (Fig 3F). On the other hand, for grandparents in a typical three-generation household, the risk of infection from inside the household was twice the risk from outside.

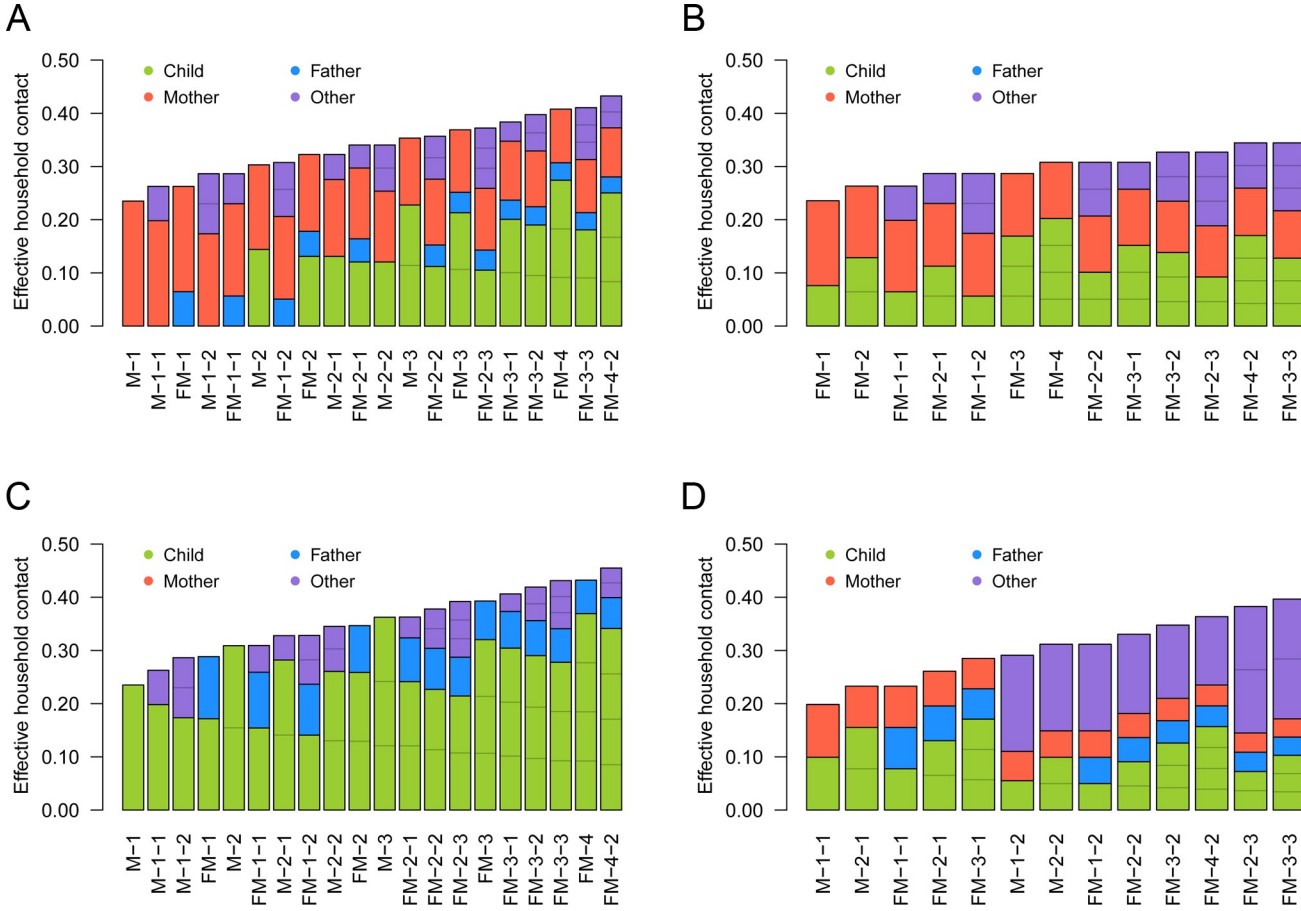

**Fig 4.** The effective amount of contacts experienced by individuals ($\eta_{kl}$) in different household compositions. (A) Child; (B) Father; (C) Mother; (D) Other. The coloured compartments denote the breakdown of effective contacts allocated to each individual in the household, which corresponds to SITP given that individual is infectious.

Once influenza was brought into a household by a student, the conditional risk of infection in other members of the household increased substantially; the implication of disease introduction into households can be seen in the simulated risk of infection after introduction (Fig 3G–3I). In "nuclear family" and "three-generation family" models, the risk in adults increased by a factor of 2–3 if a primary school student in the family was infected.

The effective household contacts that each type of individual experiences are displayed in Fig 4, indicating the substantial variation in household contact patterns between individuals and between households. SITP typically ranged around 5–20%, depending on the contact pair and household composition. Reflecting the estimated value of $\gamma = 0.5$ (CrI: 0.3–0.7), the total amount of effective household contacts was greater in larger households, but the weight of each single contact (the effective contact corresponding to contact with one individual in the household) decreased with household size. This is because the effective household contact $\eta_{kl}$ that one experiences followed an "inverse square root law", i.e., $\eta_{kl}$ is inversely proportional to the square root of the total amount of contact $C_k$ ($\eta_{kl} \propto 1/C_k^{0.5}$; see Eq 4).

Although Fig 4 summarises the heterogeneous within-household transmission patterns, one must note that the secondary transmission is conditional to infection in the primary case. When the contacts were weighted by the risk of external infection to visualise the source of primary and secondary infections for each individual, it can be seen that the children were

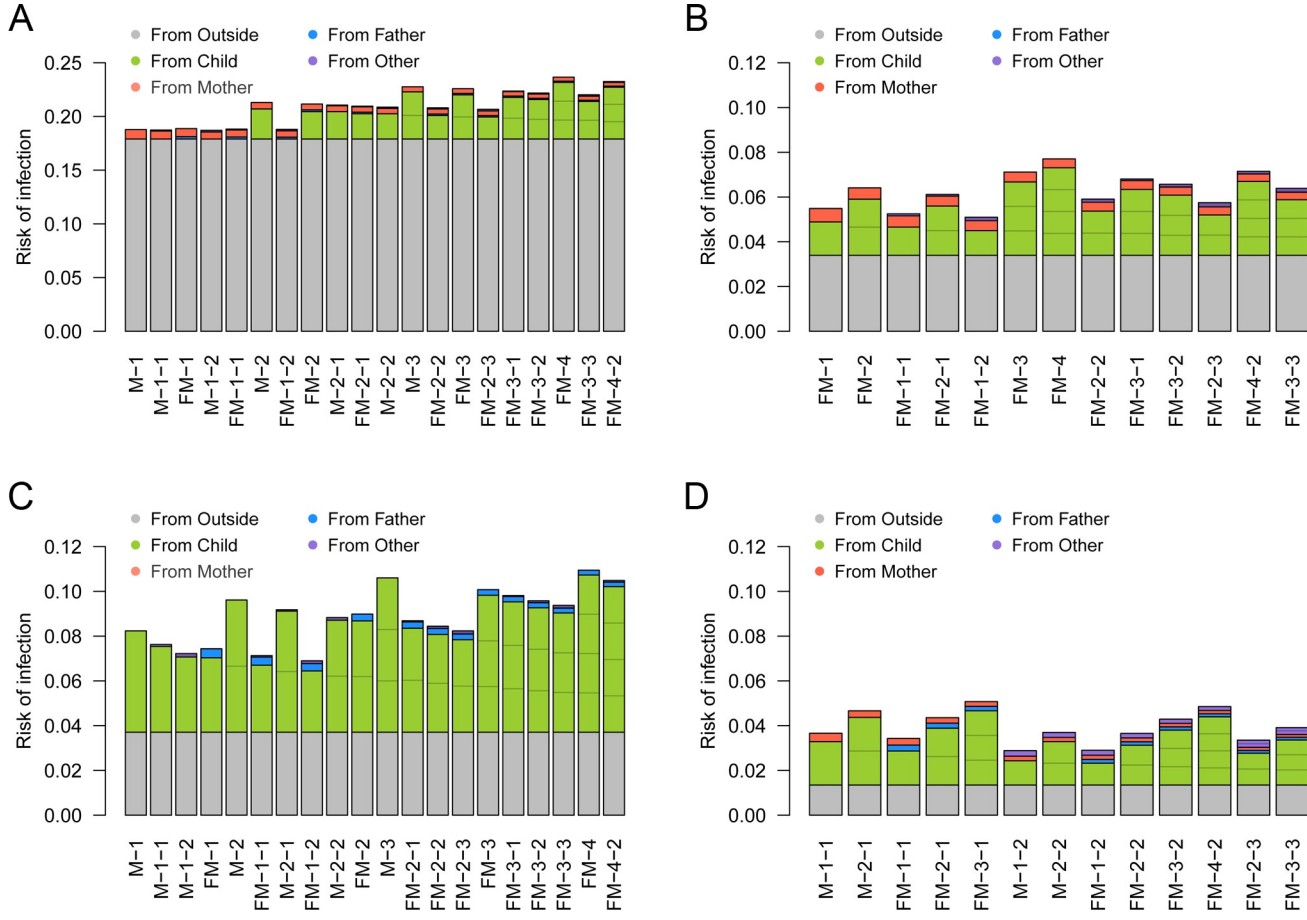

**Fig 5. The risk of primary/secondary infection to individuals in different household compositions and its source.** (A) Child; (B) Father; (C) Mother; (D) Other. The coloured compartments denote the breakdown of sources. Household compositions are displayed in the same order as Fig 4. The risk of primary infection in children was set to be 16.4%, the average between those of "students" and "siblings". Note that the scale of the y-axis in (A) is different from the other three panels.

responsible for the most of secondary transmissions within households (Fig 5): as children were more than five times likely to acquire influenza from outside the household than adults, they were the most likely source of secondary transmission. As a consequence, the individual risk of infection was mostly determined the number of children in the household. A sensitivity analysis suggested that the effective household contacts between children may have been lower than the baseline estimates under some assumptions (Figure S1 in S1 File). However, the over-all trend did not change substantially. The importance of children introducing influenza into household remained unchanged throughout the sensitivity analysis. The model prediction was highly consistent with the observed outcome patterns (Figures S2 and S3 in S1 File), suggesting our model could successfully capture the heterogeneous transmission patterns of influenza in households.

## Discussion

We applied a household-based mathematical model to a large-scale influenza survey data including 10,000 primary school students and their families in Matsumoto city, Japan, 2014–15. With the dataset of an extensive sample size on morbidity and familial roles of household

members, the model captured heterogeneous transmission patterns in households in greater detail than previous household studies.

Our results are supportive of the common perception that influenza is brought into households by schoolchildren [41]. With their high probability of contracting influenza outside the household, they were responsible for most secondary transmissions within households. Once they brought the virus from outside the household, their mother and other siblings were exposed to a higher risk of within-household secondary transmission. The estimated breakdown of infection source showed that within-household transmission accounted for a large proportion of the overall risk in adults. The relative importance of within-household transmission was especially highlighted in grandparents in "three-generation" households. In a typical three-generation family composed of two children, two parents and two grandparents, the risk of infection in grandparents was tripled by within-household transmission. Besides, it must be noted that infection of a grandparent is likely to be followed by that of another due to a high transmission risk between grandparents. These emphasise the importance of controlling school epidemic and household contagion, as the symptoms of influenza tend to be more severe in the elderly [41–43].

The results of the present study could have implications for household-level control measures. There are two steps in a household outbreak: introduction and within-household transmission. Due to the different risk patterns between the two steps, the focus of prevention measures should also change accordingly. At the pre-introduction stage when no one in the household is yet infected with influenza, the primary target is to prevent the first infection in the household from happening. Children, with the risk of external infection up to 20%, are most likely to be the first case in the household and thus should be prioritised at this stage. As the high risk of external infection is probably from schools [3], household members are advised to monitor the trend of school outbreaks and guide children to comply with daily precautions [44,45]. Our results suggest that vaccinating children is an effective strategy not only because their risk of infection is high but also because they are responsible for a substantial fraction of within-household secondary infections. Especially for adults living with many children, protecting children from infection is as important as (or even more important in some cases) protecting themselves. If one of the household members contracts influenza despite the pre-introduction control effort, the primary target shifts to preventing further transmissions within the household. Household members are now exposed to an infectious person within the same household, which substantially elevates their risk. At this post-introduction stage, preventing subsequent transmissions is important because every additional infection further increases the exposure. Our findings on household transmission patterns can be used to identify key individuals in the household network. For example, if the primary case is a child, the most probable secondary case is either the mother or another sibling. If the mother gets infected, that may be followed by transmission to either the father or another child. Direct transmissions between children and father/grandparent may be relatively rare. Grandparents are suggested to be at lower risk of infection from other household members. However, their contacts with each other are closer than any other pair of household members, which warrants attention provided the high disease burden of influenza in the elderly.

To our best knowledge, the present study first reported a parametric relationship between within-household influenza transmission and household composition with high precision. With a detailed dataset consisting of up to 10,000 households, the present study was able to employ a highly flexible modelling framework to explore previously used modelling assumptions in great detail. A decrease of the per-person risk of within-household infections with household size has been observed in previous studies [9]; our model selection supported that this reduced effect of household contact is better characterised as a function of the total

amount of contact experienced by an individual ($C_k$) rather than the household size ($N$) and that the relationship follows an inverse square root law. Previous modelling studies used different frameworks to study the relationship between SITP and household composition. Cauchemez et al. (2004) and (2014) [15,46] selected the frequency-dependent mixing assumption (SITP inversely proportional to $N$) over the density-dependent mixing (SITP independent of $N$). Many similar studies were also supportive of the frequency-dependent mixing assumption [14,19,23], while Azman et al. (2013) reported an increased transmission rate in larger households (SITP proportional to $N^{0.7}$; although not conclusive due to the limited sample size). One of the strengths of our results is that not only did we propose a better alternative measure to scale SITP than household size, we also differentiated the model from both density- and frequency-dependent models with sufficient support. The best model suggested that within-household transmission patterns lie half-way between the two extremes of density- and frequency-dependent models (we call this the semi-density-dependent model as the total effective contact experienced by an individual is proportional to the total contact intensity to the power of 0.5). Although a similar approach (without incorporating heterogeneous contact patterns) was employed in [19], where the authors estimated the STIP proportional to $1/N^{1.2}$, their CrI was too wide (0.13–2.3) to be conclusive. The large-scale dataset enabled us to obtain a narrower CrI (0.30–0.72) that distinguished the model with significance from the density- and frequency-dependent models. In the semi-density-dependent model, the total amount of effective contact increases in larger household despite the reduced importance of each contact (Fig 4). Therefore, if the risk of external infection is similar between household members, having many household members is a risk factor (which is not usually the case in the frequency-dependent model) because the effect of reduced SITP is outweighed by the increased number of household members who potentially bring infection into the household. Although such effect was not clearly visible in the present study due to the almost exclusive primary infections in children (Fig 5), more distinct characteristics may be seen in other epidemic settings with the semi-density dependent model.

Multiple limitations in the present study must be acknowledged. Firstly, the case definition in the dataset was not very strict. The data was collected by self-written questionnaires and it was impossible to validate their response. In the dataset, all student cases were reported to be with a clinical diagnosis, and more than 95% of diagnoses were based on rapid diagnostic tests [30]. Considering that primary school students in Japan are highly motivated to visit medical institutions to obtain a leave of absence from school, we believe that our data was able to capture influenza incidence in primary schools at high accuracy. However, it is not clear if the same applies to their household members; diagnoses were not explicitly required for household members on the question sheet, although the term "influenza" rather than "influenza-like illness" was used. Moreover, subclinical infections were probably present both in children and adults. Because of this, we considered underreporting in the sensitivity analysis, leaving the main conclusions unaltered. Secondly, our model formulation is only one possible candidate for parameterising within-household transmission patterns. "Contact" in our model was merely a hypothetical quantity and may not be directly related to actual physical or social contacts. We also had to use a relatively simple contact pattern matrix for successful parameter estimation. Although our model successfully explained the current data incorporating in an interpretable manner, future development may include theoretical frameworks that can explain empirical household contact patterns. A recent study have suggested the possible age-dependency in the contact frequency between siblings [7], but the age of household members were not available in the current dataset. More informative dataset and understanding of age-dependent household contact patterns will yield further clarification on this point. Furthermore, one must be aware that our analysis based on a unique study population, i.e., households

with at least one primary school student in Matsumoto city, may not be overgeneralized. Extrapolating our household transmission model to household compositions not included in the dataset, e.g., households with no children, may be unreliable. Thirdly, the present study radically simplified the risk factors of individuals. Covariates other than familial roles and household compositions, e.g., comorbidities, vaccination history, previous exposures or habits of personal hygiene, were not considered. The risk of external infection in children was estimated as a single value, which may potentially vary between classes, grades and schools. Overdispersion in infectiousness as addressed in [14,47,48] was also assumed to be negligible. Nonetheless, it is of note that the model had a fairly good performance despite considerable simplification.

Although more follow-up studies that supplement our findings are to be awaited, we believe that the present study has presented useful insights on the household-level dynamics of influenza. Understanding of the household-specific contact patterns will help us illustrate how influenza spreads across multiple social settings and facilitate individual and political decisions on disease control accounting for household-specific characteristics.

## Supporting information

**S1 File. Supplementary materials.**
(PDF)

## Author Contributions

**Conceptualization:** Akira Endo.

**Data curation:** Mitsuo Uchida.

**Formal analysis:** Akira Endo.

**Methodology:** Akira Endo, Adam J. Kucharski, Sebastian Funk.

**Software:** Akira Endo.

**Supervision:** Adam J. Kucharski, Sebastian Funk.

**Visualization:** Akira Endo.

**Writing – original draft:** Akira Endo.

**Writing – review & editing:** Mitsuo Uchida, Adam J. Kucharski, Sebastian Funk.

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
