## [Decision Letter · Decision Letter 0]

20 Oct 2019

Dear Dr Endo,

Thank you very much for submitting your manuscript, 'Fine-scale family structure shapes influenza transmission risk in households: insights from primary schools in Matsumoto city, 2014/15.', to PLOS Computational Biology. As with all papers submitted to the journal, yours was fully evaluated by the PLOS Computational Biology editorial team, and in this case, by independent peer reviewers. The reviewers enjoyed your study and the detailed influenza dataset you analyzed, but identified some aspects of the manuscript that should be improved.

We would therefore like to ask you to modify the manuscript according to the review recommendations before we can consider your manuscript for acceptance. Your revisions should address the specific points made by each reviewer and we encourage you to respond to particular issues Please note while forming your response, if your article is accepted, you may have the opportunity to make the peer review history publicly available. The record will include editor decision letters (with reviews) and your responses to reviewer comments. If eligible, we will contact you to opt in or out.raised.

- Supporting Information uploaded as separate files, titled 'Dataset', 'Figure', 'Table', 'Text', 'Protocol', 'Audio', or 'Video'.

We hope to receive your revised manuscript within the next 30 days. If you anticipate any delay in its return, we ask that you let us know the expected resubmission date by email at ploscompbiol@plos.org.

Sincerely,

Cecile Viboud

Associate Editor

PLOS Computational Biology

Rob De Boer

Deputy Editor

PLOS Computational Biology

[LINK]

Reviewer's Responses to Questions

**Comments to the Authors:**

Reviewer #1: The comments are uploaded as an attachment

Reviewer #2: Review of Endo et al

I would like to start by asking whether you may accept my apologies for submitting the review late. I hope I have not inconvenienced the authors or editors too much.

Endo and colleagues have written a thorough analysis of a very impressive dataset, of self-reported influenza infections from ~10 000 households in Matsumoto, a lovely city in Japan nestled in the northern alps. The dataset is to-die-for: 10 000 (/13 000) primary school students’ parents reported the household structure is captured to a fairly high resolution—though duplicate entries may have arisen and there is some difficulty in enumerating siblings—and whether they were infected by influenza over the course of a season. The authors note that Japanese doctors often test for influenza infection with a rapid diagnostic test (which is true, though I have no idea why they do this) which minimises the diagnostic bias that comes from requiring self-reporting. They use these data to fit models that let them distinguish the differential rates of transmission between agent types and to identify that the contact rate lies somewhere between density and frequency dependence. These are interesting findings that advance our knowledge of influenza transmission in households. I have some suggestions for improvement in the presentation of the results that I would recommend the authors consider.

My primary concern is why are the estimated infection risks from outside the household so close to the final attack rates observed empirically? ~20% of students were infected from outside vs 20.5% of students being infected in total. ~16% of siblings were infected from outside vs 18% infected in total. (Etc) This suggests that households are almost wholy irrelevant for influenza transmission, which does not accord with the authors’ own description of the importance of households for transmission or with the findings of other studies. Is the distribution of final attack sizes in their data close to binomially distributed? This should follow from these transmission rates, and would be a good empirical test of validity of the finding and estimates.

Of course the other way odd results could arise would be due to a coding mistake. Have the authors simulated a fake dataset (with substantial within household transmission) and shown that the routine can recover the parameters and hence behaviour used?

My other main critique is that the authors don’t really demonstrate the validity of their models except through a non-intuitive model fit section in the appendix. This merely compares the proportions of households with a particular configuration in model and data, but doesn’t show the distributions of configurations directly so does not allow the reader to assess whether there are systematic biases as might be caused by inability to capture the correlation structure correctly. Also the way they present these obfuscate the comparison because there is no like-for-like comparison.

Instead I would strongly recommend the authors present final attack size distributions for households of varying sizes (probably summing over configurations within a size) as this will clearly and directly show whether the model and data accord.

It is great that the authors provide code on github. I would invite them to think also of what kinds of output from their analysis might other researchers want to use as inputs into other papers, and consider whether such output can also be provided? For instance, one would expect that transmission risk parameters as a function of ego and household composition would be very useful for other simulation models: would the authors consider providing this (or other output?) in a user-friendly way?

How tenable is the assumption that there is no transmission between households? Clearly it is not in reality true, but how much does ignoring this invalidate the likelihood?

Minor comments:

Please clarify equation 1: does epsilon apply ONCE per season or is it applied once each generation of infection? Logically it must be the former, but my interpretation of the equation (which may be wrong!) led me to think that it may be the latter. Please clarify.

The authors should mention in their ethics statement that the original study was approved by a Japanese IRB.

Table 2 probably could appear in the appendix as it is rather dull compared to the other exciting parts of the paper.

Line 132: presumably it was not primary school students but their parents who did the reporting?

Line 153 (and elsewhere): best not to start sentences with lower case letters, even if they are Greek. Can this be reworded to avoid this? Eg The risk of external infection, epsilon, for each type of etc…

Line 173: This section may be better before Line 145 as it helps clarify the entries in N and n.

Line 241: Define ESS on first usage. (Currently defined on second usage…)

Table 3: clarify why beta is not sampled. Presumably because it is a derived quantity?

Line 299: The issue of not capturing multiple events among siblings should be addressable directly through the likelihood function, using the property exemplified in the following case. If X=(x1,x2,x3,x4) is MNM with parameters n and p=(p1,p2,p3,p4) then X’=(x1,x2,x3+x4) is MNM with parameters n and p’=(p1,p2,p3+p4).

Line 466: Surely it goes without saying that subclinical infections were present!

The authors could remove a few abbreviations: RDK, WBIC, CPI are scarcely used.

Figure 2: Consider rotating these panels by 90 degrees to allow the labels currently on the x axis to appear horizontally on the y axis. The CIs on panel B come out hard to see in a B&W printer. Consider modifying the colour.

Figure 3+: Ditto, consider rotation to improve readability

Appendix after equation S1: Were there really no same-sex parents in >10 000 households? Isn’t Matsumoto quite bohemian given the density of artists there?

Appendix S3: what is the subscript in line 2 of ascertainment bias?

Appendix S3ii: What are children adults? (Sounds like a rival to glay)

Finally the authors should give both the main manuscript and the supplement a more careful going over: there are some English lapses in the former and quite a few in the latter. As I understand, the supplement is not copy edited at all, so the authors should be quite careful lest their work appear sloppy.

**Have all data underlying the figures and results presented in the manuscript been provided?**

Reviewer #1: Yes

Reviewer #2: Yes

PLOS authors have the option to publish the peer review history of their article (what does this mean?). If published, this will include your full peer review and any attached files.

Reviewer #1: No

Reviewer #2: No

---

## [Editor Report · Decision Letter 1]

8 Dec 2019

Dear Dr Endo,

We are pleased to inform you that your manuscript 'Fine-scale family structure shapes influenza transmission risk in households: insights from primary schools in Matsumoto city, 2014/15.' has been provisionally accepted for publication in PLOS Computational Biology.

In the meantime, please log into Editorial Manager at https://www.editorialmanager.com/pcompbiol/, click the "Update My Information" link at the top of the page, and update your user information to ensure an efficient production and billing process.

One of the goals of PLOS is to make science accessible to educators and the public. PLOS staff issue occasional press releases and make early versions of PLOS Computational Biology articles available to science writers and journalists. PLOS staff also collaborate with Communication and Public Information Offices and would be happy to work with the relevant people at your institution or funding agency. If your institution or funding agency is interested in promoting your findings, please ask them to coordinate their releases with PLOS (contact ploscompbiol@plos.org).

Thank you again for supporting Open Access publishing. We look forward to publishing your paper in PLOS Computational Biology.

Sincerely,

Cecile Viboud

Associate Editor

PLOS Computational Biology

Rob De Boer

Deputy Editor

PLOS Computational Biology

---

## [Editor Report · Acceptance letter]

20 Dec 2019

PCOMPBIOL-D-19-01156R1 

Fine-scale family structure shapes influenza transmission risk in households: insights from primary schools in Matsumoto city, 2014/15.

Dear Dr Endo,

I am pleased to inform you that your manuscript has been formally accepted for publication in PLOS Computational Biology. Your manuscript is now with our production department and you will be notified of the publication date in due course.

With kind regards,

Sarah Hammond
